

# Spatial heterogeneity of multi-scale trade-off synergies in ecosystem services

Qiaozhen Guo, Yaxin Tian, Yue Zhang and Yajiao Wang

School of Geology and Geomatics, Tianjin Chengjian University, Tianjin, Asia, China

## ABSTRACT

**Background:** Elucidating the intricate interrelationships among ecosystem services is an indispensable prerequisite for collaborative management of diversified ecosystem services. The interaction mechanism between services across multiple spatial dimensions provides a valuable reference for the formulation of ecological conservation strategies and territorial spatial planning policies.

**Methods:** Based on the Integrated Valuation of Ecosystem Services and Trade-Offs (InVEST) model, three ecosystem services of water yield, carbon storage, and soil conservation were evaluated in Suzhou City from 2000 to 2020, their spatial-temporal distribution characteristics were analyzed, and cold hot spots were identified. Using the difference comparison method, the spatial heterogeneity of ecosystem service trade-off synergies was analyzed from the two scales of grid and county. The scale effect was analyzed.

**Results:** The results demonstrated that water yield, soil conservation, and carbon storage first declined and subsequently increased in Suzhou City from 2000 to 2020. The spatial-temporal pattern of other ecosystem services altered dramatically, with the exception of the slight spatial change in soil conservation. The proportion of non-cold-hot spots in water yield was the highest. The proportion of soil conservation and carbon storage coldspots was the highest. At the scale of 2 and 10 km, the relationship between water yield-carbon storage and carbon storage-soil conservation was dominated by trade-off. The interaction between water yield and soil conservation was mainly synergy. But the spatial agglomeration characteristics were different at different grid scales. At the county level, there were a few minor differences in the interactions among ecosystem services. The findings of this study can be a source of direction and information for fine management strategy design and implementation as well as for the creation of an ecological planning blueprint in Suzhou City.

# INTRODUCTION

Ecosystem services are now gravely threatened by the inappropriate use and development of natural resources brought on by the social economy's ongoing and fast growth. The pattern and process of the ecosystem are changed. It negatively impacts individuals' quality of life and presents a significant threat to both regional and global environmental safety (*Tang et al., 2020*; *Zhou, Tian & Jiang, 2018*). The ecosystem services are often considered an appropriate tool to better examine and enhance regional sustainability (*Schröter et al.,*

Corresponding author
Qiaozhen Guo,
gqiaozhen@tcu.edu.cn

*2017*). Ecosystem services assessment can provide a conceptual framework for interpreting and understanding the interaction between human activities and environmental protection (*Torres, Tiwari & Atkinson, 2021*; *Kirby, Zawadzka & Scott, 2024*). Consequently, a thorough understanding of the complex connections among ecosystem services is essential for improving land use planning and developing effective ecological restoration strategies (*Liao, Li & Liu, 2024*).

Studies related to ecosystem service assessment mainly focus on quantifying ecosystem service (*Fu & Yan, 2023*), analyzing the interaction relationship among ecosystem service (*Shen et al., 2023*; *Tu, Cai & Liu, 2023*; *Yuan et al., 2024*), exploring the driving factors of ecosystem service change (*Xia, Yuan & Prishchepov, 2023*; *Li et al., 2024c*), and applying it in land planning and formulating ecological protection policies (*Sun et al., 2022*). The relationship of ecosystem service interaction is divided into trade-off and synergy. Many existing studies of ecosystem services have analyzed the interactions of ecosystem services at a single point in time, usually limited to a single spatial scale (*Ren et al., 2022*). However, an increasing number of recent research have acknowledged that ecosystem services vary in scale and throughout time (*Li et al., 2024a*). At the time scale level, the trade-off synergies of ecosystems may increase, weaken or completely change direction with the passage of time (*Zhu et al., 2021*). There is a temporal lag in the modifications to the ecosystem services process (*Zhang et al., 2020*). Consequently, the trade-off synergies of ecosystem services that can be realized in a single time point might not last long. At the spatial scale level, determining the synergistic effect of ecosystem service trade-offs at different spatial scales is considered to be the key to sustainable management of the ecological environment (*Deng et al., 2023*; *Hou et al., 2023*). Existing studies have involved social organization scale (*Wang et al., 2022*; *Bi et al., 2023*), grid scale (*Zhou et al., 2022*) and physical geographic scale (*Jiang et al., 2023*). Ecosystem services may exhibit different patterns of trade-offs and synergies at local, regional, and global scales (*Renard, Rhemtulla & Bennett, 2015*). At the local scale, urbanization and land-use changes can significantly alter the balance among ecosystem services (*Wu et al., 2019*). In contrast, at the regional scale, climate change and policy interventions may play a more dominant role in shaping these interactions (*Zhang, Shi & Tang, 2021*). Ecosystem services often exhibit significant spatial variations, which can affect the outcomes of trade-offs and synergies (*Li, Zheng & Pan, 2022*).

Ecosystem services in urban areas may face more trade-offs due to high human activity (*Chen, Deng & Xu, 2024*). Determining the features of the evolution of ecosystem services and their scaling over time is a crucial first step for decision makers when debating the trade-offs and synergies of ecosystem services. This aids in the formulation of their land use plans and the management goals for ecosystem services. This process enables decision-makers to tailor policy measures aimed at enhancing synergies between ecosystem services (*Sun et al., 2022, 2024*). Therefore, to ensure the comprehensiveness and foresight of decision-making, it is crucial to not only focus on the spatial distribution differences of ecosystem services, but also to consider their dynamic changes and manifestations across different temporal scales.

The Yangtze River Delta region occupies the core position of the national major strategic planning. Southeast of Jiangsu Province is Suzhou City, which is situated in the center area of the Yangtze River Delta. It officially entered the ranks of megacities in September 2023. Significant changes in land use brought about by fast development will also have an impact on the worth of ecosystem services. Studying the evolution trajectory of its environmental services value is hence highly representative. The aim of this study was to reveal the spatial heterogeneity of ecosystem service trade-offs synergies at multiple scales in Suzhou City, and to analyze its scale effects. In this study, the fast-reading economic development period was selected from 2000 to 2020. The Integrated Valuation of Ecosystem Services and Trade-Offs (InVEST) model was used to simulate water yield, carbon storage, and soil conservation. The spatial-temporal changes of ecosystem services were analyzed, as well as the distribution characteristics of cold and hot spots. The spatial heterogeneity and scale effects of ecosystem service trade-offs synergies at 2 km, 10 km and county scales were analyzed using the difference comparison method. This article contributes a new perspective to spatial layout planning and governance strategies. It deepens the understanding of the interaction between ecosystem services in terms of their evolution laws and multi-scale features in the spatial-temporal dimension. The study aims to provide more solid support and guidance for optimizing territorial space layout and formulating eco-friendly policies.

# MATERIALS AND METHODS

## Overview of the study area

Suzhou City is situated between 30°47′-32°02′N and 119°55′-121°20′E. There are 8,657.32 km² in total. The city is mostly plain and low to the ground. The region is home to a large number of lakes and rivers. Suzhou City occupies the majority of Taihu Lake's water surface (Fig. 1). It is a famous water town in Jiangnan. It is characterized by distinct seasonal changes, mild and pleasant climate, and abundant and evenly distributed precipitation. Suzhou City, as one of the important economic centers in China, has experienced rapid urbanization and significant land use changes in recent years. Urban expansion, industrial development and agricultural activities have a profound impact on ecosystem services, and provide a rich case study of ecosystem services trade-offs and synergies.

## Data source

The data selected include land use data and natural environment data. The main data source was determined to be land use data monitored by remote sensing. The interpretation types include six categories and 25 subcategories of land use. The digital elevation model uses newly released data from NASA. The resolution of potential evapotranspiration data and precipitation data is 1 km. The soil data uses soil data set. The specific data sources are shown in Table 1.

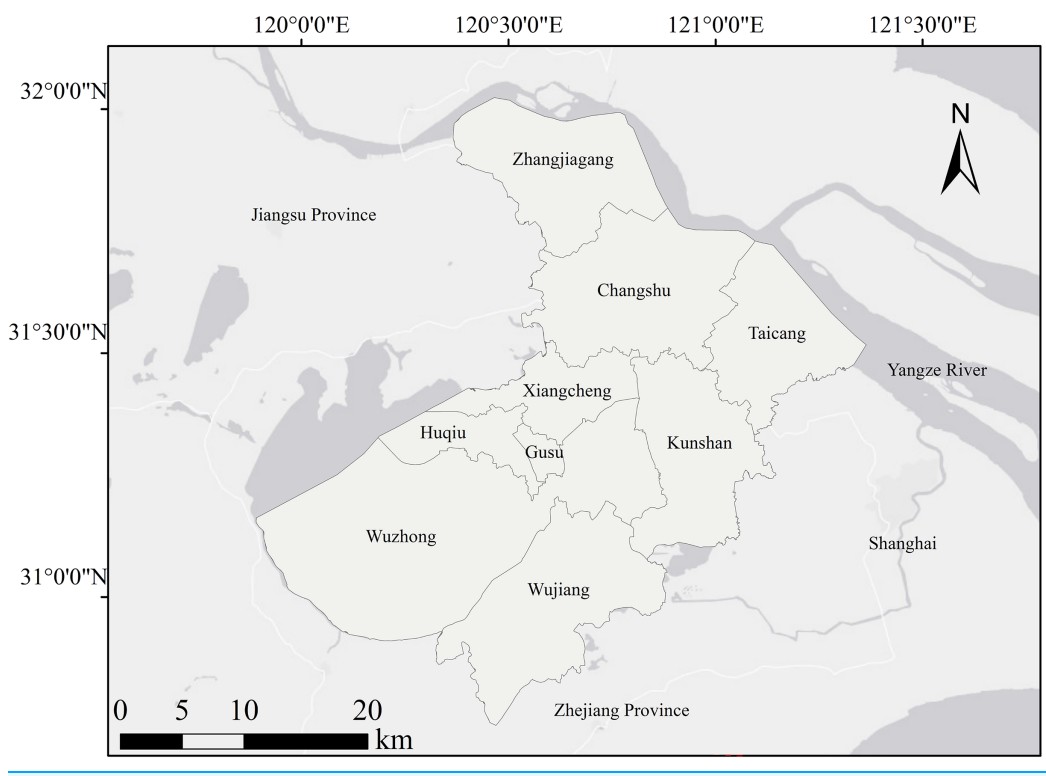

**Figure 1 Geographical location map of Suzhou City.**

**Table 1 Data sources.**

| Data type | Data sources |
| --- | --- |
| Land use data | Resources and Environmental Sciences, Chinese Academy of Sciences (https://www.resdc.cn/) |
| Digital elevation model | NASA (https://www.nasa.gov/) |
| Meteorological data | National Tibetan Plateau/Third Pole Environment Data Center (https://data.tpdc.ac.cn/) |
| Soil data | National Tibetan Plateau/Third Pole Environment Data Center (https://data.tpdc.ac.cn/) |
| Administrative boundary data | Alibaba Cloud Datav Data Visualization Flat (https://datav.aliyun.com/portal) |

## Methods

### Analysis of the characteristics of land use type transfer

The mutual transformation information of land use types in area and geographical location at the start and finish of the study can be reflected in the land use type transfer matrix (*Chen, Mao & Morrison, 2021*). The evolution trajectory of land use type area from 2000 to 2020 is obtained. The specific mathematical expression is as follows:

$$S_{ij} = \begin{vmatrix} S_{11} & S_{12} & \dots & S_{1n} \\ S_{21} & S_{22} & \dots & S_{2n} \\ \dots & \dots & \dots & \dots \\ S_{n1} & S_{n2} & \dots & S_{nn} \end{vmatrix}. \tag{1}$$
Among them, $S_{ij}$ refers to the area of land that is transformed from Class $i$ land at the beginning of the study to Class $j$ land at the end of the study. $n$ represents the number of types of land use types.

### Ecosystem services assessment

The InVEST model (intergrated valuation of ecosystem services and tradeoffs), designed to simulate alterations in ecosystem service quality and value triggered by diverse land use patterns, offers a comprehensive evaluation of various ecosystem services. In Suzhou City, given the local environmental conditions and data availability, water yield, soil conservation, and carbon storage were selected for assessment. After quantifying the three services, the partitioned statistics method is used to map them to different scales.

(1) Water yield (WY)

The data required for the operation of this module include land use data, precipitation data, evapotranspiration data and so on. WY can be quantitatively described as the following equation (*Xue et al., 2023*):

$$Y_x = \left(1 - \frac{AETx}{P_x}\right) \times P_x \tag{2}$$

where $Y_x$ is the average annual water yield depth of grid cell $x$ on land cover type $j$. $AET_x$ is the actual evapotranspiration of land cover type $j$ on grid $x$. $P_x$ is the annual precipitation on grid $x$.

The required biophysical coefficient table is shown in Table 2, *lucode* is the land use code, *kc* is used to calculate the potential evapotranspiration to modify the reference evapotranspiration, *root_depth* is the maximum root depth of land use plants, and *LULC_veg* is the code used to indicate whether the LULC class is vegetated for the purpose of AET.

(2) Carbon storage (CS)

The total carbon storage is calculated by the formula (*Zhao et al., 2023*). The data required for the operation of this module include land use data, carbon pool data. It can be quantitatively described as the following equation:

$$C_{total} = C_{above} + C_{below} + C_{soil} + C_{dead} \tag{3}$$

where $C_{total}$ represents the total carbon storage. $C_{above}$ is the aboveground carbon storage. $C_{below}$ is the underground carbon storage. $C_{soil}$ is the soil carbon storage. $C_{dead}$ is the dead organic carbon storage.

The required biophysical coefficient table is shown in Table 3, the other four types of data respectively represent different types of carbon storage.

**Table 2 Biophysical coefficient table required for water yield volume.**

| Land type | Lucode | Kc | Root_depth | LULC_veg |
|---|---|---|---|---|
| | 0 | 0 | 0 | 0 |
| Cultivated land | 1 | 0.75 | 700 | 1 |
| Forest | 2 | 1 | 7,000 | 1 |
| Grassland | 4 | 0.65 | 2,000 | 1 |
| Water | 5 | 1 | 1,300 | 0 |
| Bare land | 7 | 0.28 | 500 | 0 |
| Construction land | 8 | 0.10 | 1 | 0 |

**Table 3 Biophysical coefficient table required for carbon storage volume.**

| Land type | Lucode | C_above | C_below | C_soil | C_dead |
|---|---|---|---|---|---|
| | 0 | 0 | 0 | 0 | 0 |
| Cultivated land | 1 | 2.55 | 39.23 | 9.29 | 0.45 |
| Forest | 2 | 3.12 | 5.75 | 12.73 | 0.11 |
| Grassland | 4 | 0.09 | 0.15 | 9.97 | 0.03 |
| Water | 5 | 0.44 | 0 | 8.11 | 0 |
| Bare land | 7 | 0 | 0 | 0 | 0 |
| Construction land | 8 | 1.50 | 0.06 | 7.30 | 0 |

(3) Soil conservation (SC)

The data required for the operation of this module include land use data, precipitation erosivity data, soil erodibility data and so on. The formula for calculating the model is as follows (*Li et al., 2024b*):

$$SEDRET_x = R_x \times K_x \times LS_x \times (1 - C_x \times P_x) + SEDR_x \tag{4}$$

$$SEDR_x = SE_x \sum_{y=1}^{x=1} USLE_y \prod_{z=y+1}^{x=1} (1 - SE_z) \tag{5}$$

$$USLE_x = R_x \times K_x \times LS_x \times C_x \times P_x \tag{6}$$

where $SEDRET_x$ and $SEDR_x$ are the soil conservation and sediment retention of grid $x$. $USLE_x$ and $USLE_y$ are the actual soil erosion of the grid and its upslope grid $y$. $SE_x$ is the sediment retention efficiency of grid $x$. $R_x$, $K_x$, $LS_x$, $C_x$, and $P_x$ represent the precipitation erosivity factor, slope length factor, vegetation and management factor, and soil and water conservation measures of grid $x$, respectively.

The required biophysical coefficient table is shown in Table 4, *usle_c* is the coverage management coefficient of USLE, and *usle_p* is the practice factor supporting USLE.

### Identification of ecosystem services hotspots

According to the division method of *Qiu & Turner (2013)*, the first 20% value of water yield, carbon storage and soil conservation was defined as the coldspots. The last 20% value

**Table 4 Biophysical coefficient table required for soil conservation volume.**

| Land type | Lucode | Usle_c | Usle_p |
|---|---|---|---|
| | 0 | 0 | 0 |
| Cultivated land | 1 | 0.22 | 0.35 |
| Forest | 2 | 0.06 | 1 |
| Grassland | 4 | 0.07 | 1 |
| Water | 5 | 1 | 0 |
| Bare land | 7 | 1 | 1 |
| Construction land | 8 | 0.20 | 0 |

was defined as the hotspots. The changing law of high and low values of ecosystem services can be seen by analyzing the spatial-temporal variations of cold and hot spots.

### Trade-offs and synergies of multi-scale ecosystem services

In order to compare the scale effects of different grid unit sizes on ecosystem services, the grid-scale and county scale were selected as the research unit with reference to the construction of existing ecosystem services grid. By querying the statistical yearbook of Suzhou City, the average area of towns and villages was estimated to be 89.25 and 4.01 $km^2$, respectively. Therefore, 10 km is selected as the grid side length to fit more closely with the township scale. A total of 2 km is selected as the grid side length to fit more closely with the village scale. The different grid side lengths are set to fit with the social organization scale of the study area (*Zhou et al., 2022*) to realize the connection between ecosystem services and management decisions at different scales.

The difference comparison approach is used in this article to investigate the trade-offs and synergistic interaction of several ecosystem services (*Zhang et al., 2020*). If the multiplication of two ecosystem service changes is positive, it is a synergy relationship. If the multiplication of two ecosystem service changes is negative, it is a trade-offs relationship. The quantified ecosystem services data was mapped onto different scales. Then, the difference comparison method was used to derive the trade-offs and synergies of ecosystem services across these scales.

$$A_{T1} - A_{T2} = \Delta A$$
$$B_{T1} - B_{T2} = \Delta B$$
$$\Delta A \times \Delta B \geq 0$$
$$\Delta A \times \Delta B < 0.$$

(7)

In this expression, T1 and T2 refer to two different time phases. For ecosystem service A, its value in T1 and T2 phases is labeled $A_{T1}$ and $A_{T2}$, respectively. Accordingly, the values of ecosystem service B at T1 and T2 are recorded as $B_{T1}$ and $B_{T2}$, respectively. The variables $\Delta A$ and $\Delta B$ specifically reflect the respective magnitude of changes in ecosystem services A and B from T1 to T2.

## RESULTS AND ANALYSIS

### Changes of land use pattern

Based on the land use situation of Suzhou City, the dominant land use types were cultivated land and water, reflecting the region's low-lying river network characteristics. In 2000, cultivated land and water were the primary land uses, followed by construction land. Cultivated land is predominantly distributed from north to south, while water include the Yangtze River, Taihu Lake, Yangcheng Lake, and numerous lakes and ponds in the south. Construction land was mainly concentrated in the main urban area and surrounding counties, with forest land primarily located in the southwest (Fig. 2).

From 2000 to 2005, the most notable changes were the expansion of the urban area in Suzhou City and the growth of new and existing settlements in its northern part, leading to a significant decrease in cultivated land. Between 2005 and 2010, construction land expanded rapidly, with substantial development in both the urban area and neighboring counties. From 2010 to 2020, construction land continued to increase, but spatial pattern changes were less pronounced.

### Spatial-temporal change analysis of ecosystem service function

#### Analysis of water yield change

By using the InVEST model, the water yield, carbon storage and soil conservation were calculated from 2000 to 2020. The outcomes demonstrated that their distribution was heterogeneous. The water yield exhibited a tendency of first declining and then increasing between 2000 and 2020 (Fig. 3). The area with high water yield in 2000 was mainly distributed in the east, and the area was small. The low-value proportion reached 86.2%, mainly distributed in cultivated land and water. From 2000 to 2005, the water yield revealed a declining pattern. The area of the high-value area changed significantly. The proportion of high-value areas expanded from 1.5 to 7.3%. The low-value areas were the largest water yield types, accounting for 70.9%, which were widely distributed. Some cultivated land was also transformed into low-value areas. From 2005 to 2010, the water yield indicated a rising pattern. The area of high-value areas decreased rapidly. Some high-value areas became second-high-value. The area's percentage dropped to 0.4%, which was distributed in the eastern urban land, and the low-value area decreased rapidly to 27.0%. From 2010 to 2015, the water yield indicated a rising pattern. The area of the high-value area continued to decrease to 0.1%, mainly transferred to the second-highest value area. The proportion of the low-value area continued to decrease to 12.6%, mainly distributed around the urban area. From 2015 to 2020, the water yield showed an increasing trend. The high-value area changed from continuous decrease to rapid expansion. The area percentage of the area rose from 0.9% to 14.6%. In addition to the urban construction land, the more scattered areas in the north, east, northeast and south changed from the second-highest value to the high-value area in a cluster distribution. The area of low-value areas continued to decrease, mainly distributed around Taihu Lake, Yangcheng Lake and the southern region.
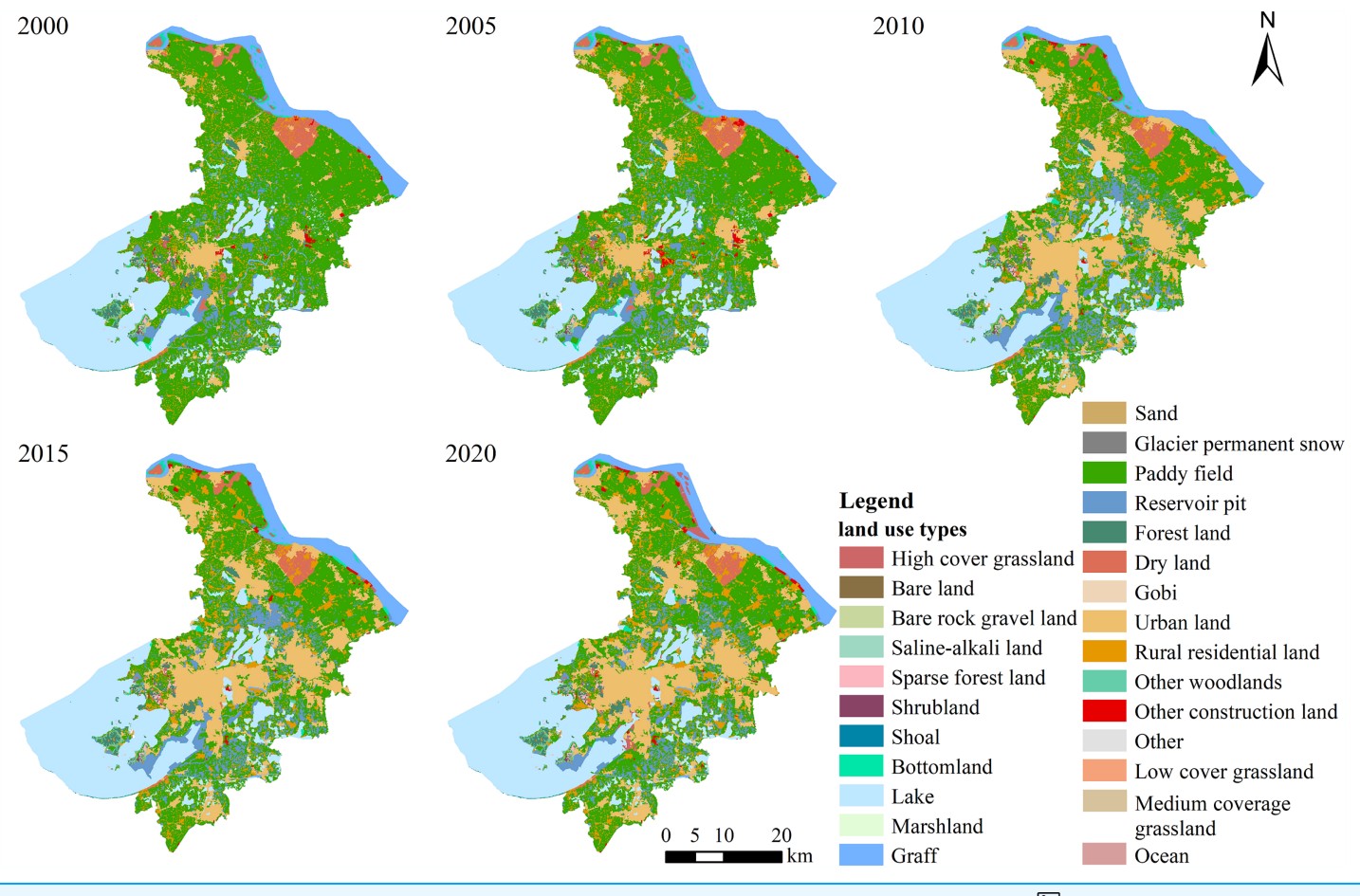

**Figure 2 Land use change in Suzhou City from 2000 to 2020.**

## Analysis of changes in carbon storage

The carbon storage demonstrated a tendency of first decline and then increase between 2000 and 2020. The distribution of the carbon storage was low in the southwest and high in the northeast in 2000 (Fig. 3). The high-value carbon storage areas made up 55.3% of the total area and were widely distributed. The low-value areas are widely distributed, including the Yangtze River, Taihu Lake, Yangcheng Lake, Chenghu Lake and many rivers and lakes. In 2005, the high-value carbon storage area underwent substantial alteration. The dilation of construction land contributed to the increase of the area of the low-value area and then encroached on the high-value area. The percentage of the area decreased to 48.4%. The percentage of low-value area increased from 42.6% to 49.8%. Combining with the above land use pattern, the surface water area indicated a slow growth trend from 2000 to 2005. Hence there was not a noticeable shift in the low-value water area. In 2010, the region of high-value carbon storage area continued to decrease, mainly turning into low-value area. The spatial pattern of the rest of the area changed significantly. Most of the high value was found in the northeast and north. The distribution was gradually fragmented due to the existence of various types. The spatial distribution of Low-value area
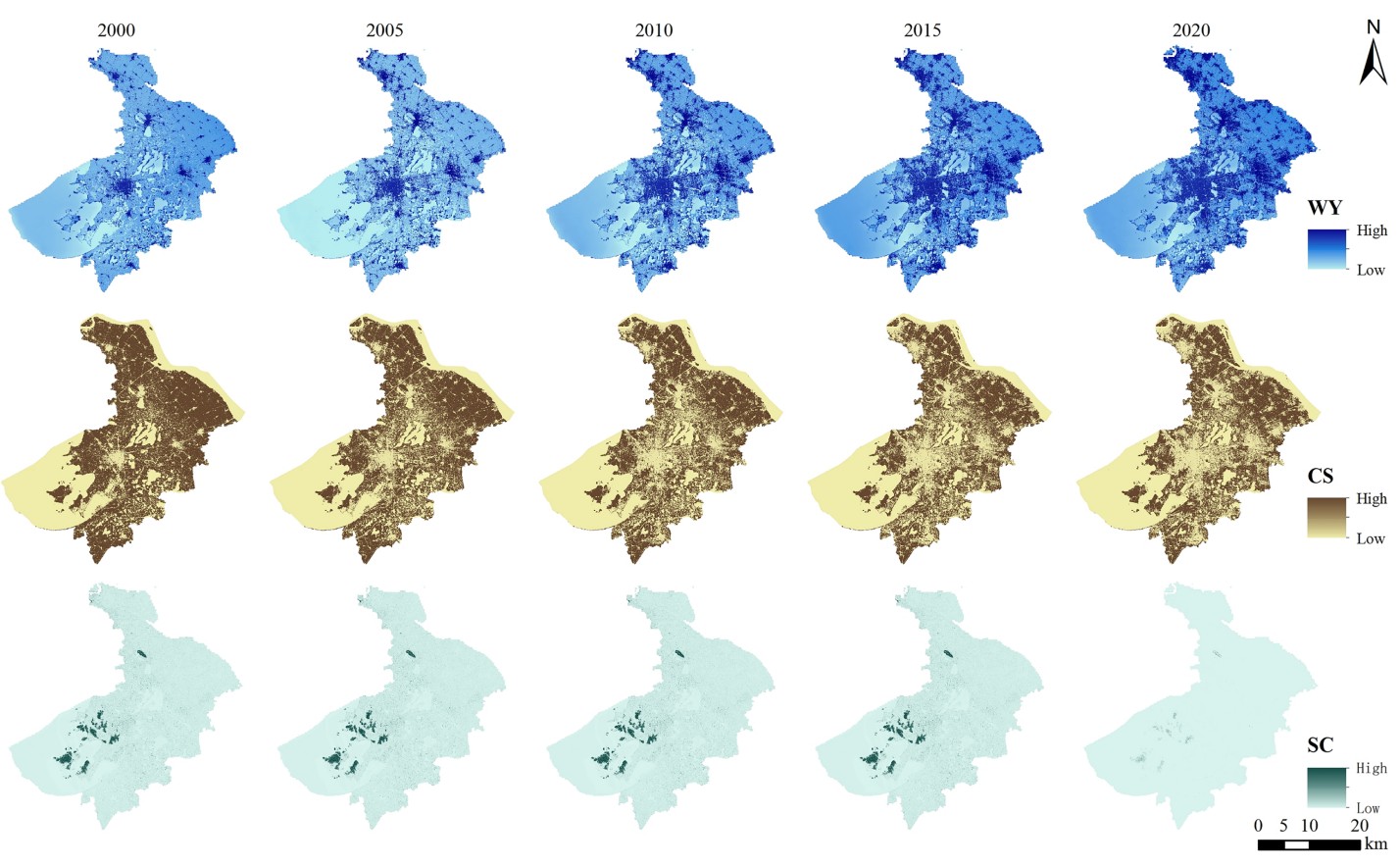

**Figure 3 Spatial distribution of ecosystem services in Suzhou City from 2000 to 2020.**

has undergone significant modification. The construction land has increased in the past five years. The proportion of low-value area increases accordingly. It is mainly reflected in the increase of urban land and rural settlements in the northern, eastern and central urban areas. In 2015, the region of the northern and eastern high-value areas continued to decrease. The proportion of the area continued to decrease to 40.8%, most of which moved to the low-value areas. The spatial distribution showed a fragmented and clumpy distribution. The area of low-value area increased slightly. The low value of carbon storage mainly included construction land and water areas. From 2010 to 2015, with the acceleration of urbanization, the area of construction land continued to increase. The water body showed a slow shrinking trend at this stage. As a result, low carbon storage value indicated an upward tendency. In 2020, the high value of carbon storage raised slightly, which was mainly reflected in the islands in Taihu Lake, the southern part of Suzhou City and the northern part of Yangcheng Lake, while the change of the remaining high-value areas of carbon storage was not significant. The proportion of low-value areas of carbon storage changed slightly, but the spatial pattern changed significantly, which was mainly reflected in the increase of the area of the median area of construction land, and the decrease of the area of the median area of waters.

*Analysis of soil conservation change*

There were two types of soil erosion that were actual soil erosion and potential soil erosion. By deducting the actual soil erosion from the actual soil erosion, one can obtain the soil conservation. From 2000 to 2020, the actual soil erosion and soil conservation first reduced, then elevated. The real soil erosion and soil conservation followed a similar spatial trend, with the distribution pattern of high values in the islands of Taihu Lake, the western part of the city and the northern part of Shanghu Lake in a large area and low values in other areas (Fig. 3). In 2000, the proportion of the area with low value of actual soil erosion was 98.6%. The proportion of the low-value area did not change significantly until 2020. The proportion of the area was always about 98.0%, and the change value of soil erosion was not significant. The spatial pattern from 2000 to 2015 is similar. Combining with the land use map, the lowest value of soil conservation is water area, construction land. The highest value of soil conservation is forest land. The other land use types are the median value. By 2020, the spatial pattern had changed significantly, some high-value areas had been transformed into medium-low value areas. And the correlation with land use distribution is less. The large range of low value distribution is the main type. There was no discernible change in the regional distribution of soil conservation between 2000 and 2020. Forest land was the highest valuable type of soil conservation. The low value was dominated by water. The soil conservation is calculated by soil erosion, and the two are closely related, rather than opposing distribution patterns. Both natural and man-made causes have an impact on actual soil erosion. The impact of human activity and soil and water conservation measures on soil erosion is mostly reflected in soil conservation.

## Spatial-temporal pattern of cold hot spots of ecosystem services

In 2000, the distribution of cold hot spots of water yield was mainly non-cold-hot spots, accounting for 62.6%. Hotspots are mainly distributed in construction land in the north, east and south. While coldspots are mainly distributed in the southwest, south and central, which is roughly consistent with the distribution pattern of water. From 2000 to 2005, the area of non-cold-hot spots decreased to 23.0%, and the area of coldspots and hotspots increased. In 2005, the distribution of cold and hot spots was dominated by coldspots, with a large area except for the eastern part of the city. The distribution location of hotspots was about the same as that in 2000, but the distribution area expanded to different degrees. Between 2005 and 2010, the regions classified as non-cold-hot spots expanded, while the areas designated as coldspots and hotspots diminished, with a significant reduction in hotspots, accounting for less than 1‰. In 2010, the distribution of cold and hot spots was dominated by coldspots, which were widely distributed in the southwest, south and central regions. From 2010 to 2015, the area of hotspots and non-cold-hot spots increased, and the area of coldspots decreased rapidly. In 2015, the distribution of cold and hot spots was mainly non-cold-hot spots. The coldspots were distributed around Yangcheng Lake, around the southern waters and some areas of Taihu Lake. The distribution area of hotspots expanded rapidly, with a wide range of distribution in the north, east and south. From 2015 to 2020, the area of non-cold-hot spots decreased, the area of hotspots increased, and the area of coldspots did not change significantly. In 2020, the distribution

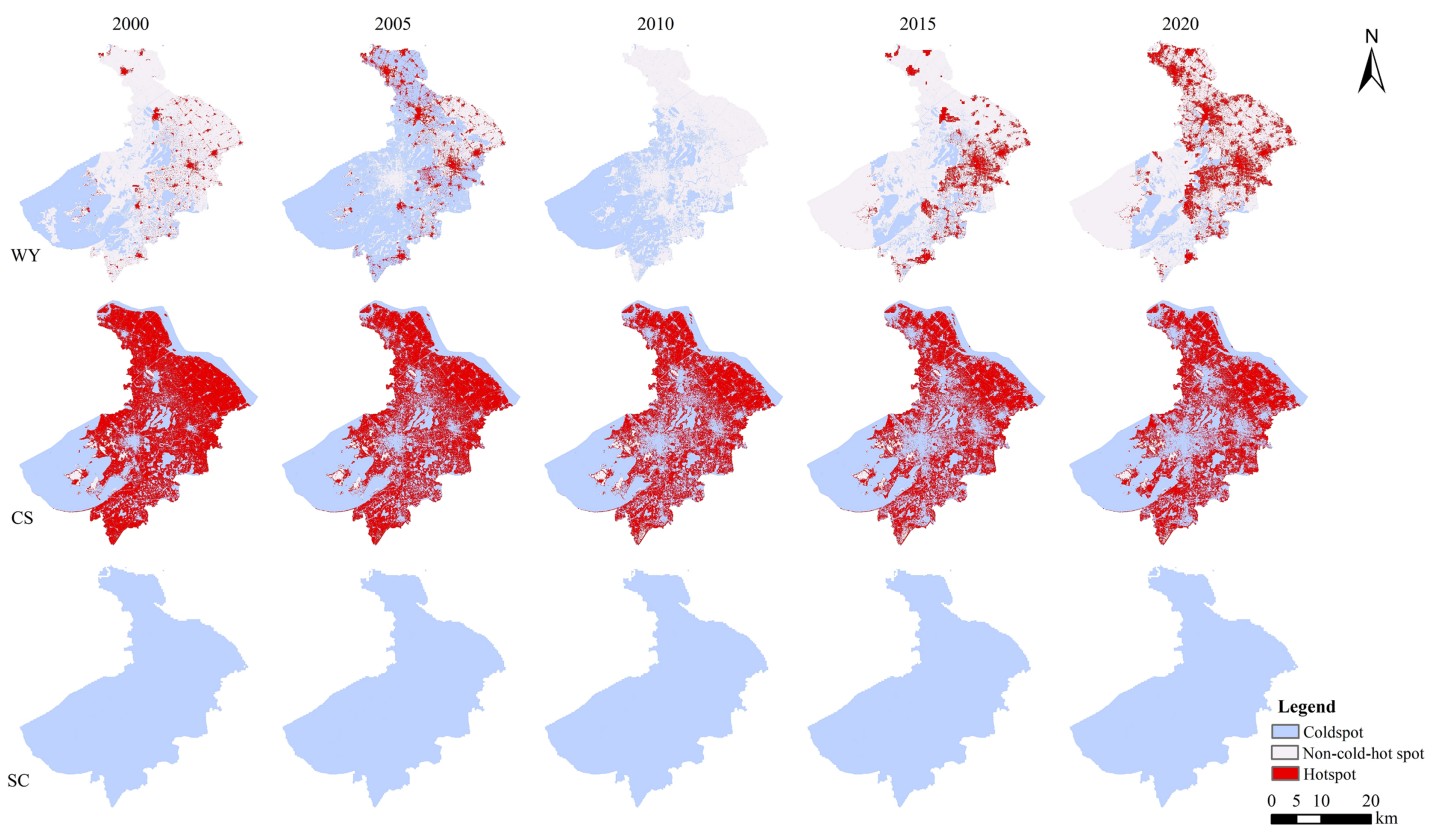

**Figure 4 Spatial-temporal distribution of cold and hot spots of ecosystem services in Suzhou City from 2000 to 2020.**

of cold and hot spots in Suzhou City was still dominated by non-cold-hot spots, and the proportion of the area decreased from 73.5% in 2015 to 68.3%. The changes of the coldspots were not significant except that Yangcheng Lake and the northern part of Yangcheng Lake evolved into non-cold hot spots and hotspots. The region of the hotspots has grown considerably, primarily located in the northern, eastern, and southern areas. Comparing with 2015, the distribution area was wider and showed a large scale distribution (Fig. 4).

From 2000 to 2020, the spatial pattern of cold and hot spots of carbon storage did not change significantly (Fig. 4). The region of coldspots exhibited a rising trend. The central distribution gradually increased. The area of hotspots as a whole showed a downward trend. The distribution was gradually fragmented. The proportion of non-cold-hot spots was always at the lowest value and were mainly distributed in the islands in Taihu Lake, the woodland on the north side of Taihu Lake and the woodland in the north of Shanghu Lake. A small range of non-cold-hot spots began to appear in the southern part of Suzhou City by 2010.

According to the classification method, the distribution of cold and hot spots obtained by soil conservation is mainly coldspots, and hotspots and non-cold-hot spots are

sporadically distributed. The spatial pattern changes are negligible between 2000 and 2020 (Fig. 4).

## Multi-scale ecosystem services trade-offs and synergies

In this research, grid scale and county scale were selected as research units. Grid scale can reflect the small changes of ecosystem services from a microscopic perspective. The county scale can be connected with decision-making management. Considering ecological process, human decision management scale and data accessibility, 2 km grid, 10 km grid and county scale were selected as research units. The spatial statistics of multi-phase ecosystem services were realized by using fishing net analysis and regional statistics tools. A total of 2 km grid, 10 km grid and county scale ecosystem services were obtained.

### Ecosystem service trade-offs and synergies in the 2 km grid

The intricate relationships among ecosystem services can be separated into trade-offs and synergies based on their respective ratios. From 2000 to 2005, the interactions between water yield-carbon storage (WY-CS), carbon storage-soil conservation (CS-SC), and water yield-soil conservation (WY-SC) of ecosystem services were mainly synergistic and distributed in a large area in the study area. Among which the synergies between WY-SC were much higher than those between WY-CS and CS-SC. From 2005 to 2010, the synergistic area ratio of WY-CS and CS-SC decreased while the trade-off area ratio increased. The trade-off was the main effect, and the synergistic effect was mainly distributed in the central urban area and the eastern region. The WY-SC was still dominated by the synergistic relationship, and the trade-off relationship was slightly distributed in the southern part of Taihu Lake. From 2010 to 2015, the proportion of trade-off area between WY-CS and CS-SC decreased. But the trade-off relationship was still dominated, and the synergies were mainly distributed around Taihu Lake and in the northeast and southeast of the urban area. The synergies between WY-SC were dominated, and a small amount of trade-offs turned into synergies during this period. From 2015 to 2020, although the regional proportion of trade-off between WY-CS continues to decrease, the dominant relationship is still in the trade-off state. In contrast, the regional proportion of the slight trade-off between CS-SC did not change much, but its spatial distribution pattern showed significant signs of adjustment, which was mainly reflected in the increase of trade-off area around Taihu Lake and the cooperative transformation of trade-off in the northeast and southeast of the city. The interaction between WY-SC is mainly synergistic, and this synergistic relationship is dominant, during this period, the trade-off effect replaced the synergistic effect in the southern portion of Taihu Lake, and the northern synergistic effect was enhanced (Fig. 5).

### Ecosystem service trade-offs and synergies in the 10 km grid

On the 10 km grid scale from 2000 to 2005, the interaction between WY-CS, WY-SC, CS-SC ecosystem services was mainly slightly synergies. The first two pairs of ecosystem services have similar spatial pattern. The spatial distribution pattern of both was mainly large area synergistic distribution, southwest of Taihu Lake was the distribution of the minor trade-off relationship. The relationship between WY-SC was dominated by a mild

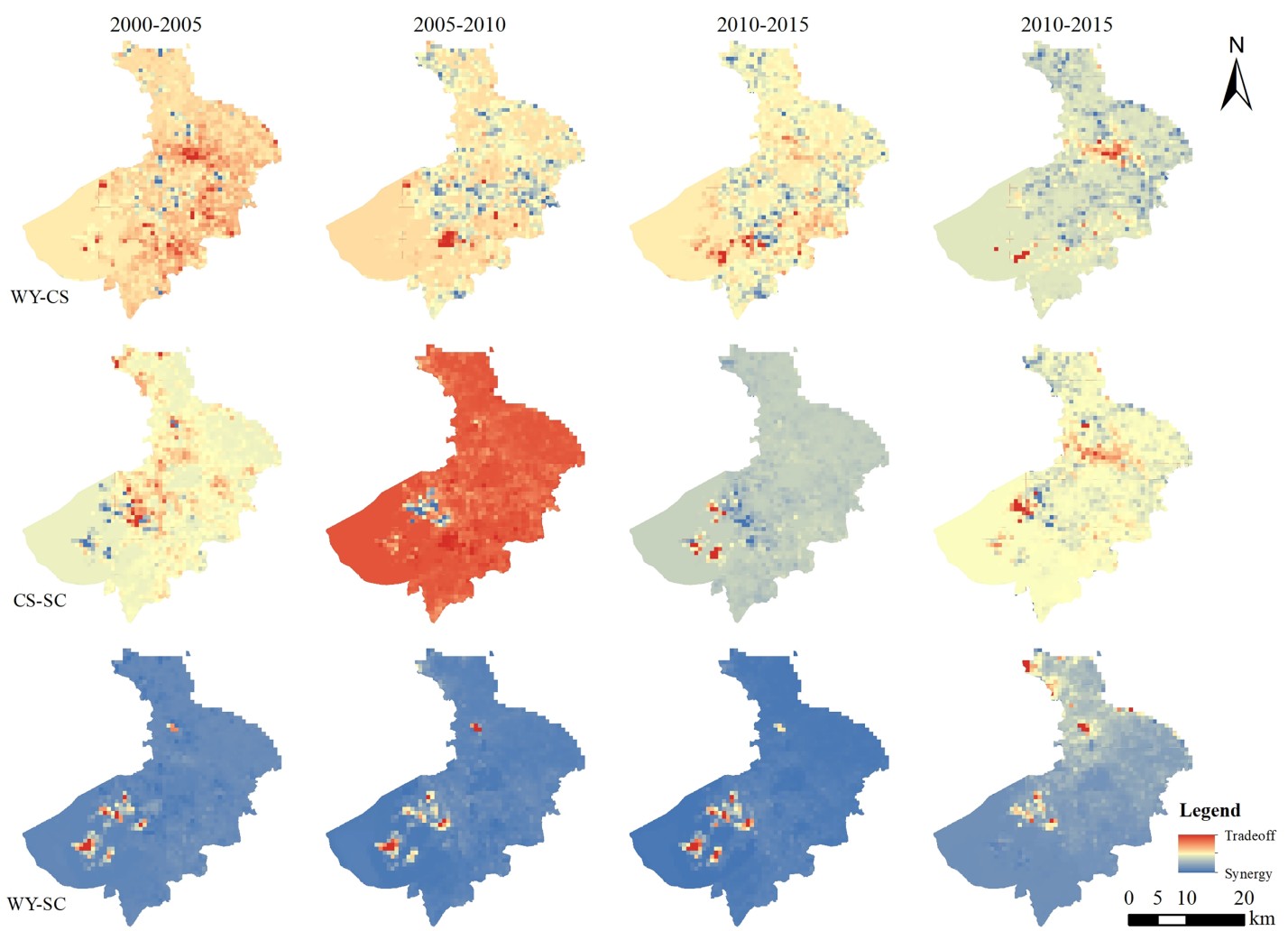

**Figure 5 Distribution of trade-offs and synergies between the 2 km grid of ecosystem services in Suzhou City from 2000 to 2020.**

synergistic effect, and the intensity of this synergistic effect significantly exceeded the synergistic effect between water WY-CS, and between CS-SC. From 2005 to 2010, there were significant changes in the relationship between WY-CS and CS-SC. In the meantime, the characteristics of the synergistic area proportion present a significant reduction, from the predominant synergistic distribution to the predominant trade-off distribution, with a large area distributed in the study area, and a small amount of synergistic relationship distributed only in the southern and central parts of the study area. The distribution of trade-off between WY-SC decreased significantly during this period. The synergistic relationship covered almost the whole study area, but the synergistic effect was weakened overall. From 2010 to 2015, the area proportion of trade-off relationships between WY-CS and CS-SC decreased. But the trade-off relationship was still dominated by the trade-off relationship. The two ecosystem service interactions have a similar spatial structure. The southwest, southeast and north had a synergistic relationship and showed an aggregation

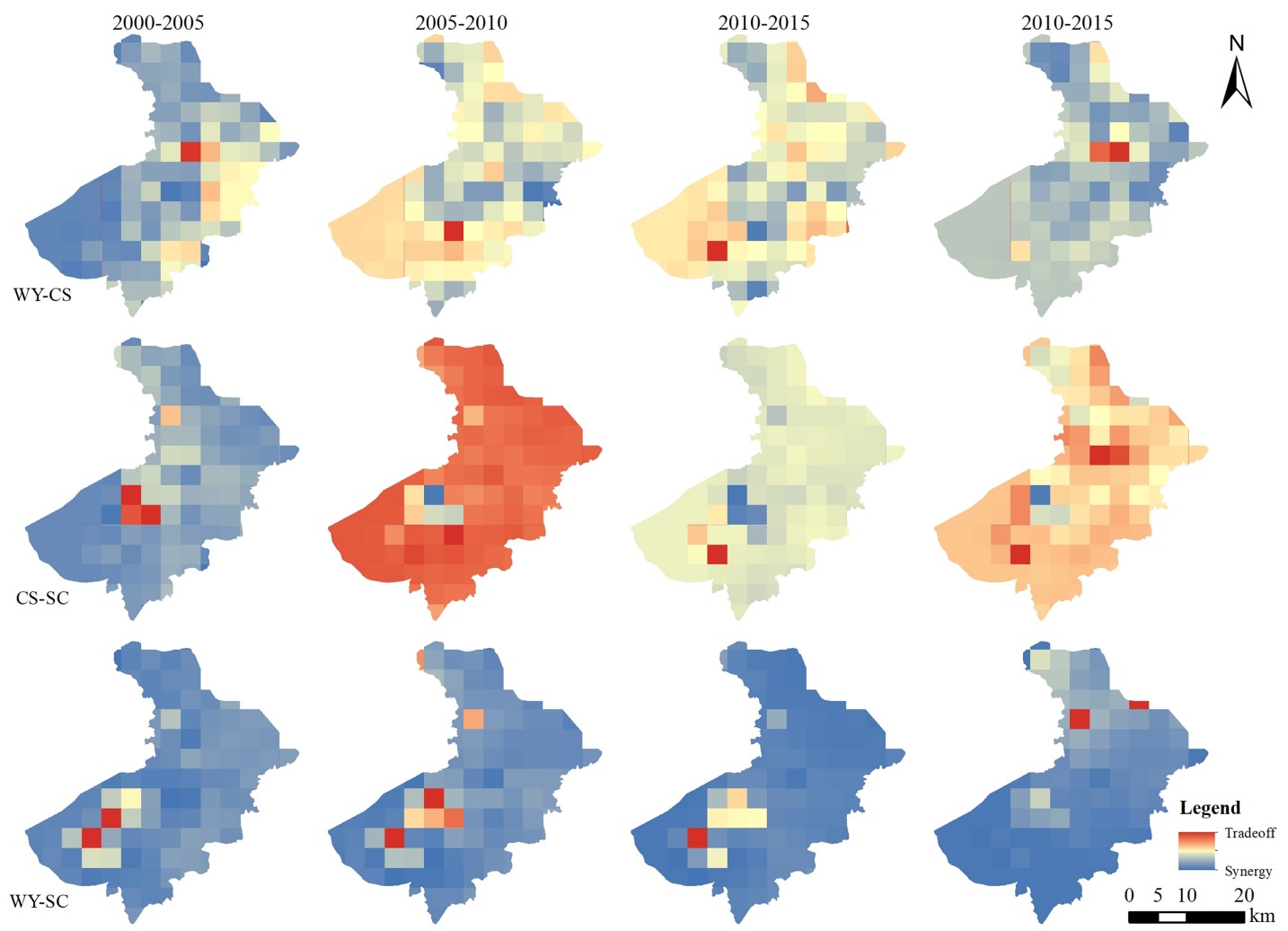

**Figure 6 Distribution of trade-offs and synergies between the 10 km grid of ecosystem services in Suzhou City from 2000 to 2020.**

state. While the rest had a large-scale distribution trade-off relationship. The distribution of WY-SC was still dominated by the synergistic relationship. But the area proportion of the middle-high synergistic relationship decreased, and the distribution area of the low synergistic relationship increased. From 2015 to 2020, the overall relationship between WY-CS and CS-SC is still dominated by trade-offs. The southwest of Taihu Lake and the south of Suzhou City suffered a marked shift in the trade-off between WY-SC. The overall function relationship is still dominated by the synergistic relationship (Fig. 6).

### Ecosystem service trade-offs and synergies at the county level

On the county scale from 2000 to 2005, the interactions between WY-SC, CS-SC, WY-CS were mainly mildly synergistic. From 2005 to 2010, the interaction between WY-CS and CS-SC changed from a synergistic relationship to a trade-off relationship. The distribution of the lowest and highest interaction between WY-CS was the same as that between 2000
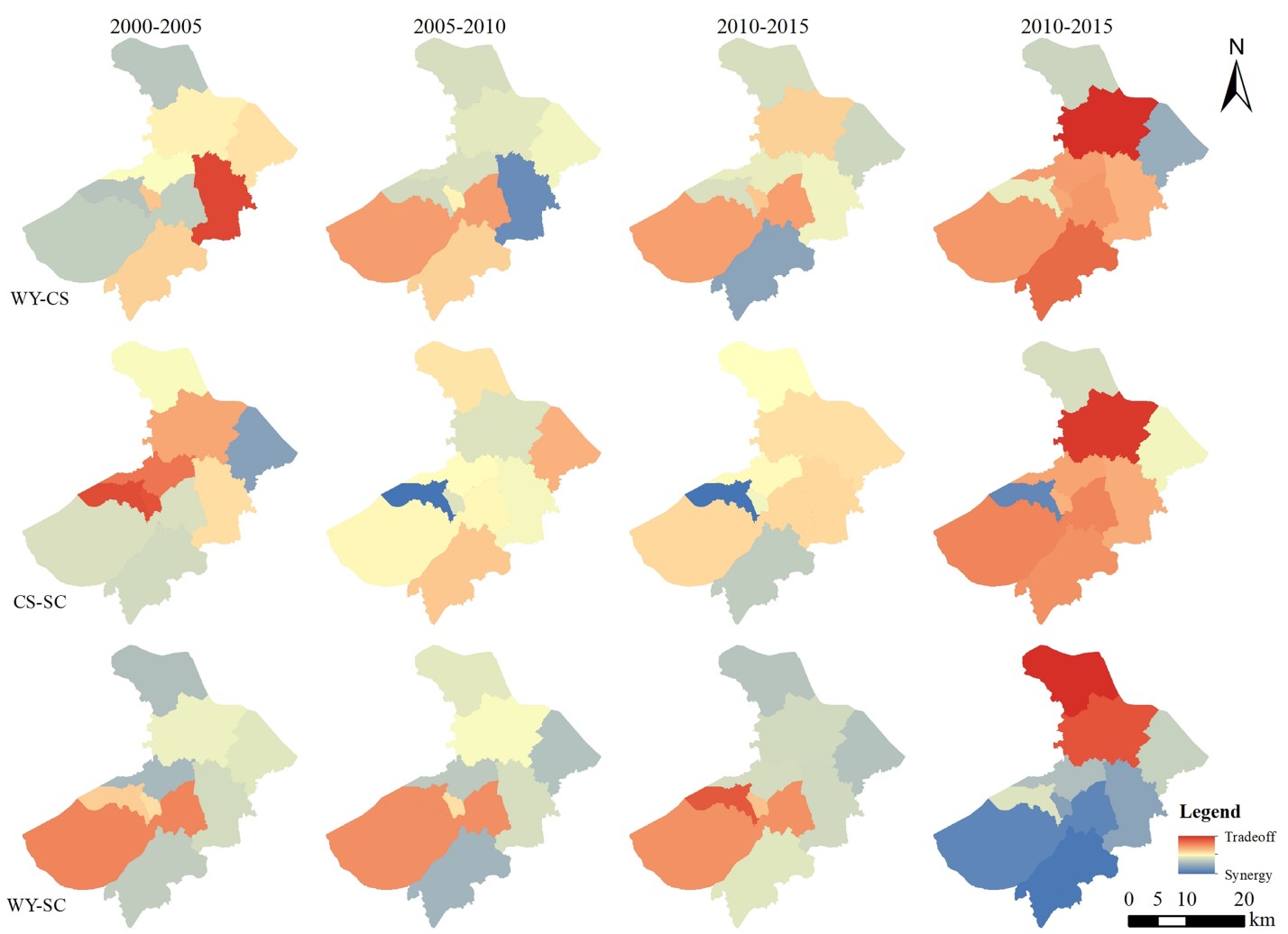

**Figure 7 Distribution of county-scale trade-offs and synergies of ecosystem services in Suzhou City from 2000 to 2020.**

and 2005. The distribution pattern was lower in southwest and higher in northeast. The distribution of high and low values of CS-SC was the same as that of 2000–2005. The trade-off effect of the two groups was not strong, and it was a mild trade-off. The effect of WY-SC was unchanged, but the synergistic effect was weakened. From 2010 to 2015, the relationship between WY-CS, WY-SC, and CS-SC remained unchanged. From 2015 to 2020, the trade-off synergies between WY-CS, CS-SC were distributed. The trade-off synergies distribution pattern was the same. Wuzhong District, Wujiang District, Xiangcheng District and Changshu City showed a synergistic effect. The other counties showed a trade-off effect, the WY-SC relationship showed a synergistic effect. The overall distribution of synergistic effects showed a pattern of weak distribution in the south and strong distribution in the north (Fig. 7).

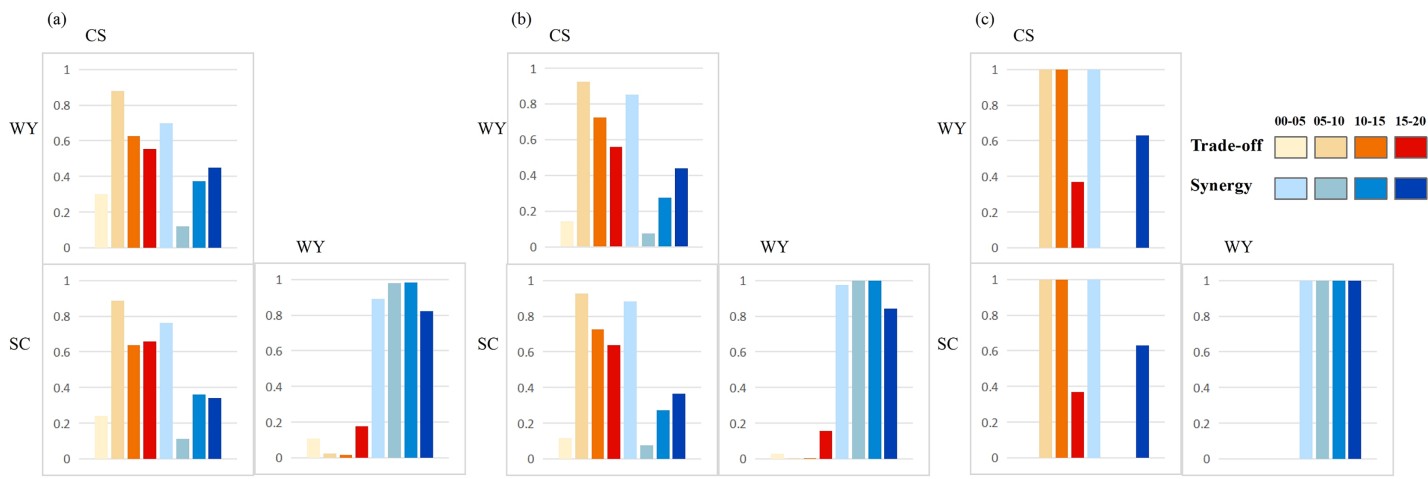

**Figure 8 Proportion of synergistic area at different scales (A refers to the 2 km scale, B refers to the 10 km scale, and C refers to the county scale).**

## DISCUSSION

### Scale effects analysis of ecosystem services trade-offs/synergies

The scale effect of the ecosystem service trade-off synergistic relationship in Suzhou City is obvious, and the ecosystem service trade-off synergistic relationship at different scales is significantly different, and even completely changes direction (Fig. 8), which is consistent with the research conclusions of other scholars (*Qiao et al., 2019*; *Xu et al., 2017*). For example, under the 2 km grid scale and 10 km grid scale, the relationship between WY-CS and CS-SC during 2015–2020 is mainly trade-off, and the relationship is mainly synergistic at county scale. This scale dependence indicates that a single-scale analysis cannot fully reflect the complex dynamics among ecosystem services. Therefore, multi-scale analysis is crucial to reveal the dynamic changes of ecosystem services, and can provide a more scientific basis for regional ecological management and policy formulation (*Gao, Hu & Liu, 2025*). One of the possible causes of the uneven synergistic relationship of ecosystem service trade-off between counties and different sizes is the spatial heterogeneity of land use type. In Suzhou City, cultivated land and water area occupy the highest proportion of land use types, accounting for 34.9–37.3% and 33.0–50.7%, respectively. A significant influence on the trade-off synergistic interaction between ecosystem services is also posed by natural environmental conditions. Geomorphic types are different in different regions. The influence of soil type, climate and other factors will also affect ecosystem services trade-off synergistic relationship, either strengthening or weakening it. How various factors affect the trade-off synergies across ecosystem services and the spatial non-stationarity responses of trade-off synergies to different factors will be the focus of further research.

## Implications for spatial management planning and ecological restoration policy making

The effects at multiple spatial scales show a strong correlation, among which the small-scale level is actually the micro continuation of the large-scale trade-offs and synergies. There are obvious differences in ecological protection and restoration strategies at different levels, which means that the applicability and effectiveness of ecological protection and restoration policies formulated on a large scale may not fully meet the unique ecological needs and constraints of small scales when they are transferred to small-scale environments (*Roces-Díaz, Díaz-Varela & Álvarez-Álvarez, 2014*). The formulation and implementation of land use and ecological protection policies should not only be confined to the boundaries of a certain administrative division, but also need to cross a single administrative unit and adopt a more comprehensive and cross-border perspective (*Chi & Ho, 2018*). Research on the complex spatial relationship between land use change and ecosystem services is helpful for the effective implementation of land use and ecological protection measures (*Kremen, 2005*).

The improvement of ecosystem services in the study area requires comprehensive consideration of their spatial-temporal heterogeneity and trade-off of synergistic relationships. From the analysis of cold hot spots, it can be concluded that carbon storage occupies a higher proportion among the multiple ecosystem service hotspots in the same grid. In view of the interaction relationship between ecosystem services in this area, the quality of cultivated land in this area can be improved by improving land use structure, optimizing agricultural industry layout, and promoting various planting methods. It is recommended to formulate reasonable management policies to further restrict human activities and prevent further degradation of the ecosystem, so as to improve the ecosystem service function.

From the perspective of county, the top three districts and counties with the lowest ecosystem trade-off synergies in Suzhou City are Huqiu District, Wujiang District and Taicang City, among which the land planning in Huqiu District was mainly ecological land and urban land, while Wujiang District and Taicang City were mainly agricultural land. The pattern of agricultural production is positioned as modern agriculture area around Taihu Lake, aquatic characteristic agriculture area and ecological agriculture area along the Yangtze River. Building green infrastructure systems (such as ecological wetlands) for their specific needs is an efficient, flexible and environmentally friendly strategy to effectively mitigate water pollution, further enhance the overall effectiveness of ecosystem services, significantly improve the natural purification capacity of water bodies, and promote the restoration of water ecosystems to a healthier state.

From the viewpoint of the grid, to guarantee the functional value of significant ecosystem services, the required ecological conservation and restoration should be carried out in accordance with the trade-offs and synergies of ecosystem services. The ecological buffer zones can be considered. In addition, the existing ecological land such as forests and grasslands should not be destroyed, the fragile ecological environment should be rebuilt in a small range. The self-healing ability of the ecosystem should be fully utilized. The service

value of the ecosystem should be restored and improved. The conditions should be created for the overall improvement of ecosystem functions and spatial planning.

## Deficiencies and prospects

This study examined how the land use pattern of Suzhou City changed between 2000 and 2020. Drawing from the analysis of heterogeneity in ecosystem services, the cold and hot spots, the trade-offs and synergies among multi-scale ecosystem services of ecosystem services were analyzed. Based on this, the article puts forward some suggestions for territorial spatial planning and ecological restoration policy in Suzhou City. As with previous studies on ecosystem services, there is no standard system to quantify ecosystem services. Therefore, this article cannot compare the analysis results with previous studies in accuracy. This study did not consider the demand for ecosystem services for the time being. The dynamic balance between the supply and demand of ecosystem services can reveal the role of ecological processes on the pattern, and is closely related to the multi-dimensional demand for ecosystem for social and economic development (*Chen et al., 2019*). In view of this, the inclusion of ecosystem service needs in the research framework can achieve more accurate and efficient decision support in the formulation and implementation of ecological conservation and restoration strategies. In addition, although the three ecosystem services selected in this article are selected from providing services and regulating services, different levels of urbanization have varied effects on environmental services. As a result, it's important to think about measuring various ecosystem services at various levels of urbanization and examining the relationships of more complex ecosystem services.

In future work, more ecosystem services should be considered for analysis at different analytical scales. More consideration should be given to the impact of urbanization and urban development (*Zhao & Li, 2020*). The evolution path and influencing factors of ecosystem services should be further understood. Consider starting from the human settlement environment, using more accurate data to exam the complex relationship and problem between ecosystem services and ecological problems on a smaller scale, and strive to improve the human settlement environment.

## CONCLUSION

Although there are some shortcomings, this research attempts to better enhance comprehension of the multi-scale implications of ecosystem service trade-off synergies. It provides a reference for the spatial planning and strategy formulation of megacities. The results are as follows:

(1) From 2000 to 2020, the cultivated land and water area were the two land use types with the largest proportion of land use. The construction land showed an increasing trend over the past 20 years.

(2) There was a trend of first falling and then increasing in the water yield, soil conservation, and carbon storage. The spatial-temporal pattern evolution characteristics of water yield and carbon storage were obvious. The soil conservation

was not obvious. The interannual variation of cold and hot spots in water yield was obvious. The whole is dominated by non-cold-hot spots. The overall carbon storage was dominated by coldspots. The proportion of coldspots increased year by year. The soil conservation was always dominated by coldspots. There was no obvious spatial pattern change.

(3) At the 2 and 10 km grid scales, there was a trade-off between WY-CS and CS-SC. The relationship between WY-SC was always synergistic during the study period. Under the two grid scales, the interaction relationship and change trend of different ecosystem services are the same, but the spatial aggregation characteristics are different. At the county scale, distinct years have distinct relationships between CS-SC and WY-CS. The spatial pattern of the relationship between WY-SC was not obvious. It was always a synergistic relationship. The synergistic effect of 10 km grid scale and county scale is stronger than that of 2 km grid scale.

### Funding
This work was supported by the Science and Technology Project of Tianjin Municipal Bureau of Planning and Natural Resources, China, grant number No. KJ[2024]21. The funders had no role in study design, data collection and analysis, decision to publish, or preparation of the manuscript.

### Grant Disclosures
The following grant information was disclosed by the authors:
Science and Technology Project of Tianjin Municipal Bureau of Planning and Natural Resources, China: KJ[2024]21.

### Competing Interests
The authors declare that they have no competing interests.

### Author Contributions
- Qiaozhen Guo conceived and designed the experiments, authored or reviewed drafts of the article, and approved the final draft.
- Yaxin Tian conceived and designed the experiments, performed the experiments, analyzed the data, prepared figures and/or tables, authored or reviewed drafts of the article, and approved the final draft.
- Yue Zhang analyzed the data, prepared figures and/or tables, and approved the final draft.
- Yajiao Wang analyzed the data, prepared figures and/or tables, and approved the final draft.

## Data Availability

The land use data and tradeoff synergy data are available in the Supplemental Files.

The China Multi-period Land Use Remote Sensing Monitoring Dataset (CNLUCC), land use data, is available in the Supplemental File and at the Resources and Environmental Sciences Data Platform: https://doi.org/10.12078/2018070201.

The digital elevation model used newly released data from NASA, ASTER GDEM Version 3, is available at: https://lpdaac.usgs.gov/products/astgtmv003.

The meteorological data and soil are available at National Tibetan Plateau/Third Pole Environment Data Center: https://data.tpdc.ac.cn.

The specific information of potential evapotranspiration data is available at: Peng, S. (2022). 1-km monthly potential evapotranspiration dataset for China (1901–2023). National Tibetan Plateau/Third Pole Environment Data Center. https://data.tpdc.ac.cn/zh-hans/data/8b11da09-1a40-4014-bd3d-2b86e6dccad4.

The precipitation data is available at Zenodo: Shouzhang Peng. (2019). High-spatial-resolution monthly precipitation dataset over China during 1901–2017 [Data set]. In Earth System Science Data (V 1.0, Vol. 11, Number 4, pp. 1931–1946). Zenodo. https://doi.org/10.5281/zenodo.3114194.

The specific information of the soil data is available at: FAO, International, I. (2019). China soil map based harmonized world soil database (HWSD) (v1.1) (2009). National Tibetan Plateau/Third Pole Environment Data Center. https://data.tpdc.ac.cn/en/data/611f7d50-b419-4d14-b4dd-4a944b141175.

The administrative boundary data is available at Alibaba Cloud Datav Data Visualization Flat:

https://geo.datav.aliyun.com/areas_v3/bound/320500_full.json.

## Supplemental Information

Supplemental information for this article can be found online at http://dx.doi.org/10.7717/peerj.19507#supplemental-information.

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
