# Peer review of "Spatial heterogeneity of multi-scale trade-off synergies in ecosystem services"

_PeerJ, doi:10.7717/peerj.19507_

## Round 0.1 · original submission · Major Revisions

The methodology section is incomplete. Key methodologies like the InVEST model must be explained in detail. Model parameters, equations, and other implementation details must be provided. Major revisions need before publication.

Reviewer 1 ·

Basic reporting

The key points of this paper are multi-scale and trade-off/synergy, and all content should be developed around these two focal points. Unfortunately, the current version is very scattered in terms of content organization.

Experimental design

The authors should clearly present the methods used to estimate the trade-offs/synergies of ecosystem services in the abstract.
The abstract includes extensive content on cold/hot spot analysis, which is inconsistent with the main topic of the title.
In the Methods section, the authors should focus on presenting the methods, steps, models, and data used in this study. There is no need to include general knowledge or introductory content.

Validity of the findings

The Discussion section should focus on discussing the methods and results of this paper. However, the current version includes a significant amount of results rather than discussion.

Additional comments

In the Introduction section, generally speaking, every sentence should have at least a citation, and these citations should be placed at the end of the sentences.
The description of the study area is too brief.
Variables in the main text should be italicized to match the formatting of the variables in the equations.

·

Basic reporting

This study utilizes the InVEST model to assess three types of ecosystem services in Suzhou, China, and attempts to compare the trade-offs and synergies across different scales. The article demonstrates a relatively complete structure and normative writing, but there are still several issues to be addressed before it can be published.

First, the study does not explicitly address scientific questions, making the manuscript like a report. It lacks an in-depth interpretation of the mechanisms underlying the observes.
Second, comparative studies at different scales are generally conducted over larger regions. The comparison between the county-level scale and the 10 km scale in this study appears overly coarse for a region the size of Suzhou, resulting in a limited sample size. For spatial studies, such a small sample size undermines the credibility of the comparisons.
Lastly, the paper exhibits immaturity and roughness in the preparation of figures and tables as well as content organization. For example, Figure 3 lacks units, and the font formatting is inconsistent. These issues suggest a lack of attention to detail in the preparation of the manuscript.

Experimental design

no comment

Validity of the findings

no comment

Additional comments

The introduction section is generally well-written, but the last paragraph should more clearly elaborate on the workflow of the study. It should also introduce the scientific hypothesis and the research questions the study aims to address.
The methods and materials section has significant shortcomings. It is particularly important to provide a detailed explanation of key methods, such as the application of the InVEST model. The parameters, equations, and other aspects of the model implementation need to be clearly described.
The resultssection requires condensation. The current content is overly lengthy and should focus on key findings.
The discussion section also has major issues, as it includes content that belongs in the results section, such as lines 446 to 468. The discussion largely revolves around simple data comparisons and lacks mechanistic exploration, resulting in a lack of depth.
Appendix: The supplementary material is largely unnecessary and can be omitted.

---

## Round 0.2 · Major Revisions

The methodology section is incomplete. Key methodologies like the InVEST model must be explained in detail. Model parameters, equations, and other implementation details must be provided. Major revisions needed before publication.

·

Basic reporting

Please refer to previously published articles in the journal and make sure to revise all figures and tables accordingly to meet the publication requirements.
The appendix files have not been properly revised. Please refer to the main text and ensure that the expression, formatting, and overall presentation are thoroughly polished.

Experimental design

no comment

Validity of the findings

no comment

Additional comments

no comment

---

## Round 0.3 · accepted · Accept

This revised version is suitable for publication in PeerJ.